# Management of Small WEEE: Future Directions for Australia

**Gimhan Jayasiri** [1,*] , **Sunil Herat** [2] and **Prasad Kaparaju** [3]

1   Environmental Engineering, School of Engineering and Built Environment, Nathan Campus,
    Griffith University, Brisbane, QLD 4111, Australia
2   Waste Management and Circular Economy, School of Engineering and Built Environment, Nathan Campus,
    Griffith University, Brisbane, QLD 4111, Australia; s.herat@griffith.edu.au
3   Environmental Engineering—Renewable Energy, School of Engineering and Built Environment,
    Nathan Campus, Griffith University, Brisbane, QLD 4111, Australia; p.kaparaju@griffith.edu.au
*   Correspondence: gimhan.jayasiri@griffithuni.edu.au

**Abstract:** Globally, the generation of small Waste Electrical and Electronic Equipment (WEEE) is rapidly increasing and accounted for over 30% of total E-waste in 2019. Managing this critical waste stream has proven challenging in Australia due to diverse product categories, short lifespans, and high frequency of disposal. To address the issue, a Multivocal Literature Analysis was conducted to identify prevailing situations, barriers, and prospects for a practical management framework. Findings were thematically analysed based on sustainability and circular-economy principles. The study revealed several critical obstacles, including the lack of involvement by local governments, the mixing of small WEEE with municipal waste, the absence of an established repair and reuse culture, and the limited domestic application of recovered metals. To address these issues, the study identifies the need for a dedicated co-regulatory stewardship scheme based on extended producer responsibility with eco-modulated fees and realistic scheme targets. Additionally, to accommodate the high product diversity, the expansion of treatment infrastructure is suggested while encouraging responsible consumption among customers. The findings of this investigation hold substantial value for the regulatory bodies in developing and implementing small-WEEE management schemes for Australia.

**Keywords:** circular economy; E-waste; extended producer responsibility; small WEEE; sustainability

## 1. Background

Electrical and Electronic Equipment (EEE) are products with circuitry and components that can be operated via a battery or power supply [1]. Waste Electrical and Electronic Equipment (WEEE), the discarded EEE, is a rapidly growing waste stream globally, comprising various hazardous, scarce, and precious metals. The environmental impacts are also significant due to greenhouse gas (GHG) emissions during EEE's production and disposal stages [2]. Global WEEE generation reached 53.6 Mt in 2019 with an average of 7.3 kg/capita, expected to rise to 74.7 Mt in 2030 [3]. Higher consumption rates, shorter lifespans, and reduced circularity fuelled these values, and they keep growing, indicating the need for sustainable management. However, this is not currently the case, as most will likely be disposed of, recycled, or traded uncontrolled [1,4,5].

Small WEEE is the most significant contributor to WEEE generation, accounting for over 30% of the total WEEE generated in 2019 [3]. Until now, management of small WEEE, except for mobile phones, has yet to be critically addressed, even though it is a significant waste stream. In addition, due to its nature, small WEEE is more likely to be stockpiled or disposed of via the kerbside collection systems as municipal waste.

In Australia, one of the largest WEEE generators in the world, only a minor portion of WEEE is regulated [6]. The country's legislative measures are inadequate, indicating that it is an issue that needs to be addressed quickly [7]. Even though WEEE accounts

for only 1% of the waste currently going to landfill, it proliferates considerably [8]. As shown in Figure 1, the subcategory of concern, i.e., small WEEE, is forecasted to be the most significant portion of the WEEE generators in the country [9], which also have the highest metal percentage among the major product categories, highlighting the importance of inhouse metal recovery [10].

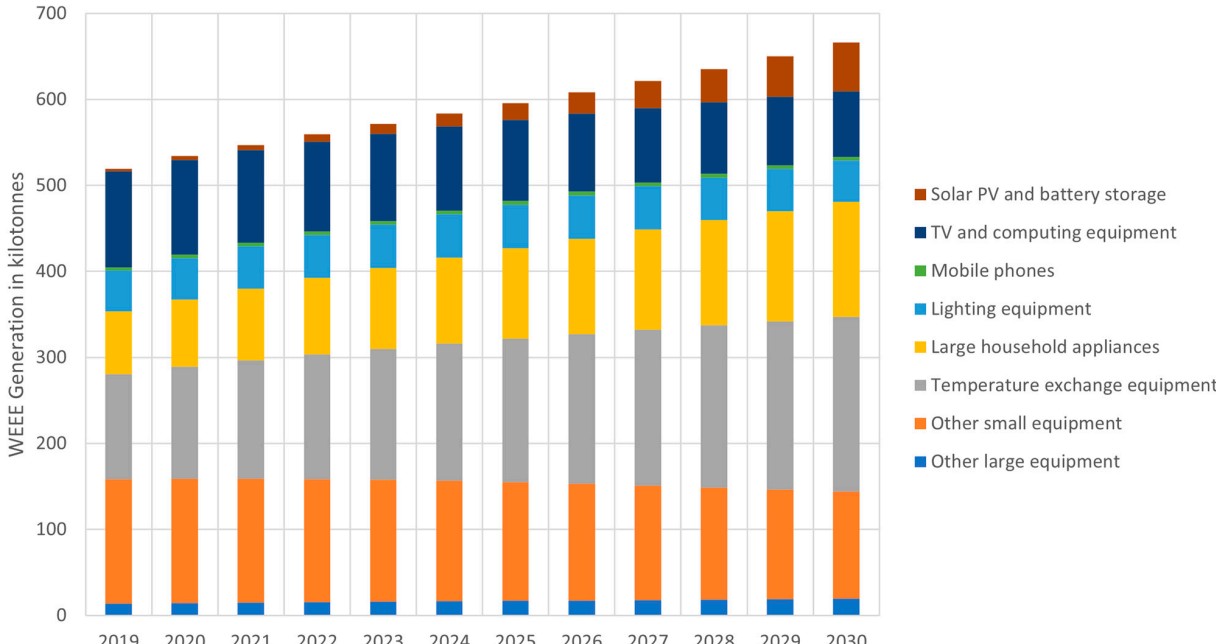

**Figure 1.** Estimation of E-waste generation in Australia [11].

Hence, a concerted effort from all the relevant stakeholders based on the sustainability concepts is essential to address this escalating crisis aligning with the Sustainable Development Goals (SDGs), where global development is growing to end poverty, protect planetary health, and ensure prosperity for future generations. WEEE management closely relates to SDGs 11 (Sustainable Cities and Communities) and 12 (Responsible Consumption and Production), and if managed properly, it will create opportunities to move towards smart cities [3]. Hence, this detailed study is of timely importance to identify the issues and prospective solutions for small-WEEE management in Australia.

## 2. What Is Small WEEE?

WEEE comprises various equipment and appliances, which must be classified to ensure consistency in management practices. Generally, products are categorised by similar function, material composition, and related end-of-life uses. In addition, homogenous average weights and lifetime are other characteristics to be considered. Under this basis, the United Nations University (UNU) has developed a classification known as UNU keys [3]. This also includes all possible EEE, which can be linked with the harmonised statistical (HS) codes. Currently, these codes are used globally, which assists in collecting statistical data on EEE in the market [12]. In addition, the classification aligns with the European Union WEEE Directive 2012/19/EU, which was amended by Directive 2018/849, narrowing down to six categories of WEEE from the earlier ten categories [12]. Those categories are:

- Large equipment;
- Temperature exchange equipment;
- Small equipment;
- Small IT and telecommunication equipment;
- Screens, monitors, and equipment containing screens having a surface greater than 100 cm$^2$;
- Lamps.

According to the global data-reporting mechanisms, small equipment is identified as small WEEE after its disposal, usually including most small household appliances, toys, video players, cameras, and power tools. However, due to different country-based management schemes, small WEEE can comprise various EEE that belongs to other UNU keys.

In the case of Australia, due to the existence of different stewardship schemes, WEEE is classified into eight main categories according to the recent discussion paper published by the Department of Agriculture, Water and the Environment (DAWE) (now known as the Department of Climate Change, Energy, the Environment and Water), namely:

- Solar PV and battery storage;
- TV and computing equipment;
- Mobile phones;
- Lighting equipment;
- Large household appliances;
- Temperature exchange equipment;
- Other small equipment (considered as small WEEE in this study);
- Other large equipment.

This list has created separate categories to manage the WEEE under the National Television and Computer Recycling Scheme (NTCRS). Mobile Phones are also considered separately, managed by a voluntary stewardship scheme: Mobile Muster. Furthermore, based on the minister's priority list, solar PV and storage were also categorised separately [13]. The list of subcategories, relevant UNU keys, and some example EEE that falls under small WEEE in Australia are given in Table 1 below.

**Table 1.** Related products identified as small WEEE in Australia [13].

| UNU Key | Description | Examples in Each Subcategory |
|---|---|---|
| 0114 | Microwaves | Microwave ovens used for domestic purposes |
| 0201 | Other small household equipment | Electric blankets, roof fans, personal weighing machines, household sewing machines, domestic appliances with electric motors, electric irons, watches, clocks |
| 0202 | Equipment for food preparation | Domestic food grinders, mixers, juice extractors, toasters |
| 0203 | Hot-water preparation | Electric instantaneous heaters or storage water heaters, electric coffee or tea makers |
| 0204 | Vacuum cleaners | Vacuum cleaners with and without self-contained motor |
| 0205 | Personal care equipment | Hair dryers, dressing, drying apparatus, electric shavers, hand-drying apparatus |
| 0401 | Small consumer electronic | Microphones and stands, headphones and earphones, and combined microphone/speaker sets |
| 0402 | Portable audio and video devices | Radio broadcast receivers that can be operated without an external power source |
| 0403 | Music instruments, radio, and hi-fi | Office machines (banknote dispensers, coin-sorting machines), stapling devices, amplifiers, sound-recording apparatus, radio and broadcast receivers which require an external power source, musical instruments |
| 0404 | Video devices and projectors | Video recorders, DVD, Blu-ray setup boxes, projectors |
| 0405 | Speakers | Single and multiple mounter loudspeakers |
| 0406 | Cameras | Television and digital cameras, video camera recorders |
| 0601 | Household tools | Drills, saws, high-pressure cleaners, lawnmowers, soldering machines |
| 0701 | Toys | Car racing sets, electric trains, music toys, biking computers, drones |
| 0702 | Game consoles | Video game consoles |
| 0801 | Household medical equipment | Hearing aids |
| 0901 | Household monitoring and control equipment | Alarms, heat and smoke sensors, instruments to measure flow level, pressure voltage, current resistance |

## 3. Materials and Methods

Management of WEEE is a widely discussed theme in the global literature. However, the management of small WEEE needs to be more critically assessed as it significantly contributes to global WEEE flows. Given this background, the present study identified and reviewed international and Australian small-WEEE management practices. A Multivocal Literature Analysis (MLA) was carried out to capture the views of practitioners and academia by referring to both formally published literature and 1st- and 2nd-tier grey literature. This methodology is commonly adapted in other fields like management and educational research to identify emerging areas and research topics apart from the knowledge in academic settings [14].

The study began with searching the online databases Scopus and Web of Sciences by employing the research query (TITLE-ABS-KEY ("Waste") AND TITLE-ABS-KEY ("small EEE") OR TITLE-ABS-KEY ("small household appliances") OR TITLE-ABS-KEY ("small WEEE") OR TITLE-ABS-KEY ("small Equipment")). The initial search yielded 71 and 189 articles from Scopus and Web of Sciences databases within the past two decades. Then, these articles were carefully reviewed to identify the pieces with a focus on policy and management under the pillar of WEEE. The selected papers were primarily concentrated in the Journal of Cleaner Production, Waste Management, and Resources, Conservation and Recycling. Then, a Google Advanced Search was employed to identify relevant government documents, reports, and discussion papers to obtain the grey literature. This resulted in 120,000 hits, and suitable documents were selected based on the page rank algorithm of Google. When choosing the appropriate grey literature for this study, the producer's authority and objectivity were considered in the exclusion criteria to maintain the quality. The finalised set of the literature included 202 documents, as shown in Figure 2 below.

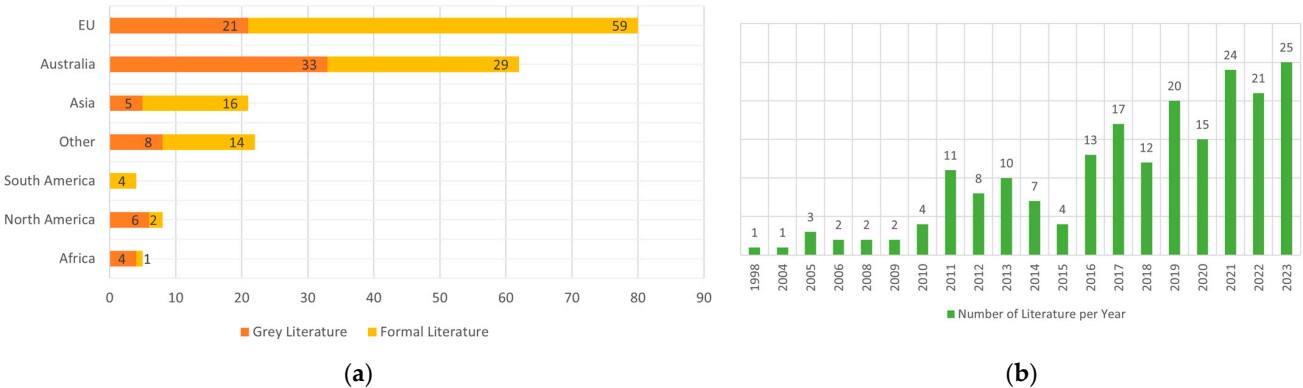

(**a**) (**b**)

**Figure 2.** Details of relevant documents finalised from MLA. (**a**) Regional breakdown of referred literature. (**b**) Number of items of the literature based on publication year.

The analysis of the documents revealed that direct reference to small WEEE is meagre, and most of the related discussions are part of a broader topic focusing on WEEE. Given this background, the results and discussion were broken down into three sub-sections, as explained below.

- Global scenario: An overview of global regulations, policies, and frameworks was discussed with reference to small WEEE. An explicit focus was given on Japan, which has a dedicated law for small-WEEE management.
- Australian scenario: This section includes a detailed assessment of existing laws, frameworks, and stewardship schemes related to WEEE in Australia to identify pros and cons, which is useful for developing a framework for small WEEE.
- Progress of small-WEEE management in Australia: This section discusses current steps taken by the different stakeholders in Australia to manage small WEEE.

The information summarised in the above three chapters provided recommendations and prospects for managing small WEEE for Australia in the last chapter.

## 4. Global Scenario

Currently, most developed nations are equipped with a national register, enhanced collection systems, and detailed logistic management for WEEE [5]. European countries like Switzerland, France, and Spain are progressing in overall WEEE management while updating the regulations to match the increasing WEEE flows. However, according to the authors' best knowledge, the Japanese Small Home Appliances Recycling Law (SHARL) is the only country-wide dedicated law to manage small WEEE.

### 4.1. Japanese Small Home Appliances Recycling Law (SHARL)

Japan, one of the first countries globally to introduce an EPR-based WEEE management system, currently has two fundamental laws to ensure proper management of WEEE. The Home Appliance Recycling Law (HARL), which only targets washing machines, air conditioners, and TVs, transfers the responsibility to the consumer to pay for the collection and transportation and return WEEE to the correct recycling route [5]. In addition, Japan introduced the Small Home Appliance Recycling Law in 2013 to promote the recycling of small WEEE. The main objective of this law was to improve the collection and recycling rates of small WEEE. The current collection rate is more than 90%, which has also increased the material recovery rates significantly since 2013 [15]. However, the law differs from EPR as manufacturers only try to reduce costs by improving product designs [16–18]. The role of the stakeholders under SHARL is described in Table 2 below.

**Table 2.** Role of stakeholders in SHARL.

| Stakeholder | Responsibility/Task |
|---|---|
| Manufacturer | Should maintain ingenuity in designing the small EEE while reducing the cost by using recycled materials. |
| Retailers | Should ensure appropriate disposal of small EEE by the consumers. |
| Municipal Council | Collect information and promote research and development activities, carry out education campaigns, secure necessary funds, take necessary measures, and cooperate with retailers to separate and collect small WEE and dispose of them to certified recyclers. |
| Consumer | Should ensure responsible disposal via relevant collection methods. |
| Certified Recycling Companies | A plan should exist for the recycling business, cover the municipalities with the planned collection area, and acquire proper certifications from the government. |
| Government | Issue relevant certifications for recycling companies, provide guidance, and carry out on-site inspections. |

In parallel to enacting this law, the Japanese Ministry of Environment released several explanatory materials to increase the collection rates. A dedicated mark for recycling small EEE introduced by the government has to be used by agencies involved in the collection as an anti-theft measure [19]. SHARL also enables the protection of personal information contained in small WEEE. Adequate measures are taken to make consumers aware that this information should be erased before the disposal of products. If this is not feasible for the consumers, retailers may perform advanced operations to delete this information. Customers are also provided with recycling information, encouraging them to dispose properly [20]. Several collection strategies are recommended under this law, as stated in Table 3. In 2020, 36% of small WEEE was collected using the pickup method, followed by recovering at collection stations and the bringing-in method [21].

The collected small WEEE is then disassembled, crushed, sorted by metal type and plastic, and recycled as metal resources by metal smelters. Hazardous substances are also included in the process [19].

**Table 3.** Different collection strategies suggested under SHARL [20].

| Collection Strategy | Overview |
|---|---|
| Box method | Municipal councils (MCs) permanently set up collection boxes like public facilities, supermarkets, and home appliance stores at various locations. This is more convenient for consumers even though administration costs are higher for the MCs. Boxes should be fully protected to avoid theft. The population density should be considered when installing boxes. |
| Recover at collection stations | Existing municipal waste collection stations are upgraded with the small-WEEE category, and instructors are employed to ensure proper disposal. It is physically easy to discharge for the consumers, and the theft risk is low. |
| Pickup method | Collections according to the conventional sorting classification of general waste, such as non-burnable garbage, oversized garbage, and small WEEE, are picked up from the conveyor belt or the platform before the processing section. Sorting efficiency is low, and additional administrative costs will occur due to the vast diversity of the products. |
| Participatory collection | Citizen groups already collecting recyclables can expand the items by including small WEEE. This is convenient for customers. However, the groups and consumers should be well educated about the products accepted under the law. |
| Event collection | This is a combination of the box method and participatory collection at local events. Additional expenses have to be borne by the MCs. To collect items effectively, it is necessary to exhibit multiple affairs with different characteristics based on the event's content. |
| Bringing-in method | Consumers can bring the items to the recycling/incineration plant itself. This is time- and effort-consuming for customers, which may reduce the collection rates. |
| Door-to-door method | Municipal councils organise collection events periodically or after being informed by the respective village leaders. It is the most convenient method for consumers even though it is costly for the MCs. |
| In-store collection | Retailers receive used small electronic devices brought by consumers face-to-face at their stores. The possibility of theft is low, and consumers can dispose of their small WEEE when purchasing another item. Collection rates need to be more consistent in this case. |
| Return-flight method | This is a method of collecting small WEEE discarded by consumers on the return flight of the delivery by the retailer when another product is delivered to the consumer. This can be applied to most online sales to avoid free riding. |

### 4.2. Focus on Small WEEE in Other Frameworks

Overall, European countries are advancing rapidly in managing WEEE. However, when it comes to diverse waste categories like small WEEE, more efforts are needed to achieve higher efficiency levels, especially in collecting small WEEE [22]. In addition, it has also been highlighted that there is a need for a specific policy for managing small WEEE in the EU while providing incentives to the users [23]. According to the analysis, Canada was the only country other than Japan and the EU with a dedicated small-WEEE collection programme [24]. Table 4 summarises the related information about small WEEE in the global frameworks.

**Table 4.** Information on small WEEE in global frameworks.

| Directive/Framework/Programme | Information Related to Small WEEE |
|---|---|
| EU WEEE Directive (2012/19/EU) | Indicates that 70% shall be recovered while 50% should be recycled based on the average weights. Distributors to collect small WEEE (external dia. <25 mm) from the consumers, free of charge, without any obligation to buy an equivalent EEE [25]. |
| RoHS Directive (2011/65/EU) | Restrict the use of toxic substances like lead, mercury, and other persistent organic pollutants (POPs) during the manufacture of EEE, including small household appliances [26]. |
| The German Electrical and Electronic Equipment Act (ElektroG) | Small appliances are categorised by dimensions less than 50 cm [27]. |
| The VREG (Regulation on the Return, Acceptance and Disposal of Electrical and Electronic Devices) in Switzerland | This is an EPR-based scheme, and PROs are responsible for collecting small WEEE. Currently, the SENS Foundation is handling this task [28]. |

**Table 4.** *Cont.*

| Directive/Framework/Programme | Information Related to Small WEEE |
|---|---|
| Waste Electrical and Electronic Equipment (WEEE) Regulations in the UK | Retailers with EEE sales areas over 400 square metres should also accept small WEEE from private household customers freely at their retail sites [29]. |
| Canadian Electrical Stewardship Association (CESA) | Small WEEE is managed via the recycling programme ElectroRecycle in British Columbia. Some of the techniques followed by them to ensure the proper functioning of the scheme include consumer awareness, collective community events, and school partnerships. There is a specific product guide for small appliances, and the system also accepts products with batteries that are difficult to remove. However, consumers must drop off the products at the respective collection point [24,30]. |

### 4.3. Challenges and Provisions for Improvement

#### 4.3.1. Waste Generation and Collection

When a particular owner/user decides that there is no use for the EEE and gets rid of it, the EEE becomes WEEE. This will typically occur when the use of the EEE ends in terms of function or interest. If the functional use is over, then the owner can check to repair it, and if it is still not feasible, then the dysfunctional EEE will enter the waste stream, in which it will get recycled, be repurposed, or end up in landfill. On the other hand, there can be a scenario where the user runs out of interest in the EEE even though it is functioning well. This might occur due to the halt of software upgrades or new EEE with advanced technology available for a reasonable price in the market [31]. Then, the EEE will be reused or shared by someone else; if not, it will enter the waste stream. When EEE is disposed of as WEEE, coordinated collection mechanisms are needed to avoid stockpiling, which will delay the economic value of material recovery. Small EEE lasts four to eight years, while malfunction and technological obsolescence are two common reasons for replacing them [32–34]. Due to low collection rates, disposal of small WEEE in municipal facilities is common, leading to MRF contamination and increasing the fire risk [34,35]. It also causes a loss of economic value due to delayed material recovery [36,37]. Several other barriers and probable solutions were identified from the literature, as shown in Table 5.

**Table 5.** Barriers and challenges to practical WEEE collection.

| Barriers and Challenges | Probable Solutions |
|---|---|
| Sentimental attachments by consumers [38,39] and people with technical skills often try to repurpose old EEE [40], which leads to stockpiling in households [31,41]. | Encouraging disposal by making it part of the daily routine [40], providing incentives [42], conducting doorstepping campaigns and mobile collection systems [40], establishing electronic trackers [43], and intelligent waste collection boxes [44] while ensuring information security and safe data transfer mechanisms [17]. |
| Lack of knowledge in proper disposal [38,45] and selling the obsolete EEE to informal collectors [4,46]. | Conducting comprehensive education and awareness campaigns, including distribution kits [31,47–50]. |
| Dropping off at retailers is a rare event for consumers [40] and underutilised retailer drop-off options [47]. | Establishing collection points at supermarkets/shopping centres and retail shops [41,51] with adequately labelled and easily identifiable collection containers [36], which increases the proximity to collection points [52]. |
| Similar transportation efforts for both small and large WEEE [38], limited storage and technical capacity at collection centres [49]. | Improve logistics of collection mechanisms via encouraging direct feedback from customers [51], increased stakeholder collaboration via a clearing house [46,53], decentralised networks with pretreatment and sorting facilities [54], and introducing collective takeback schemes [55,56]. |
| Use of unattractive weight-based collection quotas [23] and lack of participation by the OEMs and retailers [39,55]. | Enhance financial support by tax reduction [49], advanced recycling fees (ARFs) [48] and encouraging producers' right of precedence [38]. |

#### 4.3.2. Design for Circularity

Small EEE has a diverse category of products with different designs and manufacturing techniques. These designs and techniques critically affect the end-of-life (EoL) treatment of EEE due to the requirement of dismantling before recycling. Suppose the manufacturers do not consider this fact during the design stage; in that case, the pretreatment mechanisms

will become time-consuming and cumbersome, affecting the supply of raw materials for the product. This was evident in a study conducted in Spain, as most of the toys were not designed considering the EoL aspects [57]. Hence, designing for disassembly using a modular approach will ease the tasks carried out in the pretreatment of WEEE [58]. In addition, eco-design principles [59] should also be followed, considering the environmental impacts of their products along the whole life cycle [60].

Remanufacturing is another concept that should be given due notice. Most of the smaller EEE is not remanufactured as it cannot be disassembled, or their disassembly is expensive. Furthermore, remanufactured products should be at least 25% cheaper than the new alternatives to win a good customer base. It would be profitable if active disassembly (AD) is carried out more largely for small EEE. The AD will further facilitate a step change in recycling the EoL EEE [61]. Establishing a collective takeback system will ensure a market for the remanufactured items, which is an economically viable option as some legislations, like the EU directive, mainly focus on waste avoidance. However, product design incentives will make remanufacturing activities more attractive [62]. Reuse and remanufacturing are always concerns of the producers, assuming it will affect the sales of new products, reducing their revenue and profits even though total sales might be improved through a better environmental image [38]. This is an issue which is yet to be analysed via scientific studies.

### 4.3.3. Repair and Reuse

In the concepts of circular economy, the first option to consider after collection is the share/reuse capability of small WEEE [63]; regarding potential climate impacts, it is the most preferred EoL management, followed by recycling [2]. However, a proper conceptual understanding of reuse, repair, and preparation of reuse (PfR) is required when developing a sustainable management system for small WEEE. In general, reuse is an act of using materials and objects to delay the disposal of waste [64]. From a waste management perspective, it is more commonly associated with obsolete EEE, which can be categorised as functional or dysfunctional. Functional EEE can be directly reused after some refurbishment if the users are interested. However, the potential to reuse the dysfunctional EEE could be higher, especially for items with low economic value or expired technology. While some EEE faces a rapid depreciation of its value, most of the ICT products with high economic value have the potential for reuse, given minimal efforts for repair [41]. However, dysfunctional EEE should be checked and repaired, known as preparation of reuse (PfR), before reusing them for the exact requirement or a different purpose, known as repurposing [65]. A repair process usually involves product identification, failure diagnosis, disassembly and reassembly, and replacement of spare parts to restore working condition. Several factors which hinder the promotion of repair and reuse of EEE in the global community are stated in Table 6.

**Table 6.** Barriers and challenges for promoting repair and reuse culture.

| Barriers and Challenges | Probable Solutions |
| --- | --- |
| Ineffective collection techniques [66]. | Collection points with testing mechanisms [39,67] to assess the functionality of small WEEE for repair and to preserve reusability [35,57,68,69], with access to WEEE with potential reuse [70]. |
| High repair costs due to scarcity of spare parts [27], disassembly issues, shorter lifespans [66], and lack of manufacturer guidance and support [40]. | Collaborative consumption and financial support by reducing VAT on repaired items [69,71], differentiated disposal fees [71], following standards for product quality and safety [5,72], conducting repair only by qualified technicians [55], access to repair information by amending copyright laws [73], maintaining a repair register [70], promoting competition, and improve repair centres to increase product lifespan [5,74]. |

**Table 6.** *Cont.*

| Barriers and Challenges | Probable Solutions |
|---|---|
| More focus on recycling and resource recovery over reuse by PROs [60,75]. | Consider both environmental impacts [41,76] and consumer demands when selecting the suitable mechanism, extending tender periods for recyclers and making PROs fully engage in reuse [55,77], encouraging upstream reuse to maintain quality [38] PROs to support a repair fund [78], and enhancing collaboration between stakeholders [40]. |
| Consumer behaviour due to hygiene and cleanliness concerns [73], willingness to repair [67], quality and reliability of repaired items [73], lack of knowledge about repair shops [79], and easy access to new products [40,73,79]. | Selling EEE indicators like the repairability index [80], ease of disassembly matrix [79], reuse potential indicator [81], and providing used EEE with a quality label about functionality and durability [79], guarantee mechanisms including software updates or repairs [73], education and awareness campaigns [71], and encouraging consumers to repair by themselves via repairability tools [79]. |

### 4.3.4. Recycle and Recovery

After the reusing and repairing options are not viable, recycling is the next feasible option for small WEEE. Recycling rates depend strongly on the material composition of the small WEEE [82]. Plastic is the dominant material in most appliances, generally used as insulators and central processing units [83]. In Europe, plastic recycling is one of the five priority areas alongside the recovery of critical raw materials (CRM) [84]. Approximately 2.6 million tons of WEEE plastics are produced annually in Europe, where 9% contain brominated flame retardants (BFRs). However, this figure is declining due to the stringent regulations imposed [85]. Further to the cascading effects on recycling due to low collection rates and improper disposal, Table 7 highlights the key barriers and probable solutions affecting the recycling and recovery process of small WEEE.

**Table 7.** Barriers and challenges affecting recycling efficiency.

| Barriers and Challenges | Probable Solutions |
|---|---|
| Limited official statistics [86] and more extensive volume and similar material consumption in small WEEE [13] lead to unrealistic recycling targets [84]. | Appropriate modelling [76] using accurate data and auditing processes, including both quantity and quality of recycled materials when calculating efficiency [87] and use of a product ID system to identify material content [43]. |
| Lack of infrastructure capacity to treat small WEEE [88]. | Effective communication between manufacturers and disassembly plants [43], performing characterisation investigations focusing on emerging appliances as early as possible [82], and increased competition between recyclers' materials [89]. |
| High cost of labour due to manual sorting of small WEEE due to product varieties [42]. | Using product-based sorting techniques [87], developing technical guidelines [87,90], and introducing property close separation during collection [91]. |

### 5. Australian Scenario

Australia is an OECD country with a population of 25.6 million as of 2021, mainly concentrated in urban areas. From 1988 to 2014, net imports in Australia gradually increased to 35 kg per capita [10], and due to the increase in income, the purchasing of EEE has risen considerably in Australia [48]. Small EEE being the most significant portion of the import categories [10,13], a vast sales volume was also observed in small equipment like microwave ovens, which recorded an increase in growth rate by 28% from 2004 to 2018. Continuing this trend, EEE put on the market averaged 671 kt (26.3 kg per capita) in 2019 [86]. The rise of import rates has increased WEEE generation rates, making Australia one of the largest WEEE generators, with a 21.7 kg per capita value in 2019 [3]. This value is closer to some of the other developed nations, such as the UK (23.9 kg per capita), Sweden (20.1 kg per capita), and Switzerland (23.4 kg per capita), and this figure keeps rising [10]. Furthermore, due to these high proportions, small WEEE largely contributes to GHG emissions, releasing around 1.1 Mt of $CO_2$e in 2019, as shown in Figure 3.

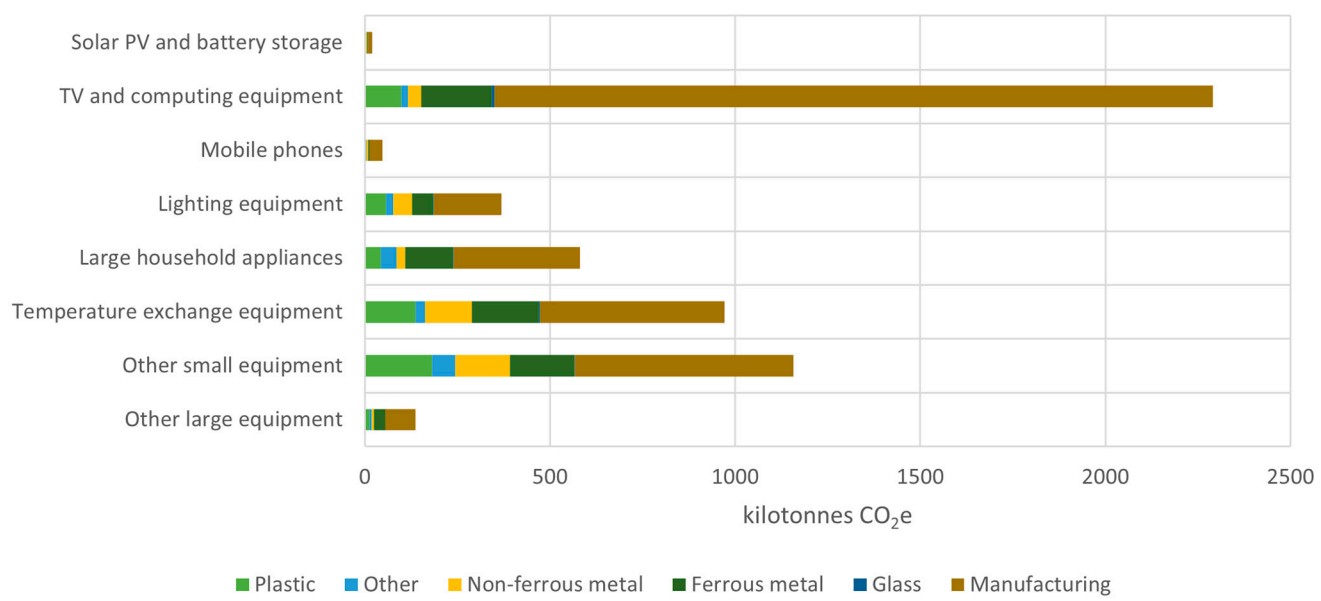

**Figure 3.** Embodied GHG emissions of WEEE in Australia in 2019 [11].

*5.1. Laws and Frameworks*

The Australian WEEE management system combines Swiss, Japanese, and other European WEEE regulations. The approaches in the system are primarily regulatory, institutional, and administrative, with a financial instrument being present to focus on the funding mechanisms [48]. During an official review in 2015, several key issues were identified in these regulations, such as limited scope for categories of WEEE, unclear roles of stakeholders, inability to provide reasonable access, lack of auditing measures, and insufficient recycling targets [7]. Given this fact, a summary of current regulatory frameworks is presented in Table 8 below.

**Table 8.** Current Australian frameworks related to WEEE.

| Law/Act/Framework | Overview |
|---|---|
| Hazardous Waste (Regulation of Exports and Imports) Act (1989) | Restricts the uncontrolled transboundary movements of waste. WEEE can be excluded provided evidence of the nonexistence of dangerous components like heavy metals and POPs [92]. |
| Product Stewardship Act (2011) | It aims to reduce waste in landfills by increased recycling and recovery mechanisms via voluntary, co-regulatory, or mandatory stewardship schemes [93]. |
| National Waste Policy (2018) | Sets out the objectives of avoiding waste, improving resource recovery, increasing the use of recycled materials in the products, managing material flows in an environmentally friendly manner, and improving data management [94]. |
| National Waste Policy Action Plan (2019) | Highlights the importance of the Product Stewardship Investment Fund to speed up the recycling schemes of EEE [95]. |
| Recycling and Waste Reduction Act (2020) | Provides a framework to manage waste products along their entire life cycle, promoting circular economy by establishing national, legislated stewardship schemes. It mandates removing plastic components and glass [96] from WEEE when exported for downstream recycling. |
| Australian Standard AS 5377 [97]: Management of Electrical and Electronic Equipment for Reuse or Recycling (2022) | Indicates guidelines and minimum requirements for proper WEEE management to maximise reuse and material recovery. Recycling service providers should be certified under this standard [97]. |

Even though no mandatory schemes exist currently to manage WEEE [96], the National Television and Computer Recycling Scheme (NTCRS) 2011 was established under the Product Stewardship Act. Being a co-regulatory scheme, it has three main objectives:

reducing WEEE sent to landfills, enhancing recovery of reusable materials in an environmentally friendly manner, and providing access to households and businesses to a recycling service. However, it provides oversight of downstream recycling despite having a higher compliance rate [98].

Except for some shortcomings, NTCRS has established a vast network for WEEE collection services across the country, free of charge for customers [99]. Due to its introduction, many waste television sets and other peripherals were diverted from landfills while applying the scheme's benefit to a broader society [100]. Under this scheme, importers/manufacturers are liable under a co-regulatory arrangement in the form of product stewardship such that they are responsible for the whole life cycle of the products. They should obtain membership of these co-regulatory bodies, i.e., producer responsibility organisations (PROs) [101].

These PROs should organise WEEE collection and recycling on their members' behalf. The collection points include permanent sites at local waste transfer stations and retail shops [102]. They must also provide independent annual reports to the government. Given the poor performance, the department taking the leading role has the authority to cancel the co-regulatory arrangement. Besides these main parties, local and territory governments also play a pivotal role in collecting WEEE from the customers, even though it is not mandatory. They also have the authority to take other measures outside the scheme.

The main stakeholders of the scheme, the consumers of EEE from households and small businesses, do not have a substantial responsibility, and due to this, the effect of free-riders has been raised [48]. Consequently, the local council's bulky household waste collection gets mixed up with WEEE like TVs, computers, and printers [96]. Apart from the NTCRS, no significant co-regulatory schemes are functioning targeting WEEE. However, several other voluntary projects like Mobile Muster, the Australian Battery Recycling Initiative, and Cartridges 4 Planet Ark focus on managing mobile phones, household batteries, and printer cartridges, respectively [99]. Furthermore, B-Cycle, another government-accredited stewardship scheme, was launched in 2022 to provide free battery recycling to consumers [103].

### 5.2. Waste Generation and Collection

In Australia, in 2014 more than 50% of the WEEE collected was used for material recovery, while a minor portion was exported for reuse [104]. Given this, one-quarter of the total WEEE generated in 2019 accounted for small WEEE due to its shorter life expectancy not exceeding five years. Due to the falls in many products like video players, a decline in the volume of small EEE put on the market is expected, representing a 15% decrease. However, the small WEEE generation in Australia is estimated to almost double by 2035. Food equipment and household tools are expected to be significant products in this category by 2030 [13]. Hence, the scope of WEEE management schemes in Australia should include small WEEE to reduce the potential environmental impact.

Looking at the generation patterns, it was noted that most of the products are thrown away due to the limited firmware updates, [6] and without clearly understanding the legislated WEEE to be recycled [7]. Under the current WEEE management system, state and local governments manage small WEEE outside the scope of NTCRS. This is comparatively different from countries like Japan and Switzerland, which have more streamlined practices. Takeback schemes are also limited in the country [7], and not all local authorities work with PROs under NTCRS [96]. However, some councils (e.g., Brisbane City Council) have made arrangements to enable the customers to drop off their small household appliances like toasters, kettles, microwaves, and vacuum cleaners at nearby retail shops like Big W, the council's resource recovery centres, and other WEEE collection services like EcoActive and Ecocyle [105]. Hence, broadening the NTCRS to include small WEEE would result in better community service.

In general, the collection rate for small WEEE is as low as 50% [11], mainly due to the customers' hoarding behaviour, which reduces the potential for resource recovery [106].

For the collection programmes to succeed, reasonable access should be maintained [107], and more than one service per 100 km in the regional areas is needed [7]. In addition, due to the consumers' absence of legal obligations, complaints related to access to collection points were also scarce [96]. Local councils are key WEEE collection channels even though they still need to be given a broad responsibility by the legal framework of the scheme [7,108]. Illegal storage of WEEE by recycling companies can also be avoided by following proper reporting, and it will help the federal government identify specific drop-off locations used only for counting [108].

### 5.3. Repair and Reuse

The "right to repair" concept empowers consumers to repair faulty EEE, allowing them to take control of their products. Given the growing demand for repair in Australia [31], there should be a greater focus on addressing the repair needs of small EEE, which is currently in need of an established repair market. During 2020–2021, 200 tonnes of EEE were reported to have been reused in South Australia, confirming the consumer willingness to repair and reuse [109]. However, compared to white goods, small EEE is more sensitive to repair costs, as the typical cost of repairing it often exceeds the price of purchasing a new replacement [6]. This situation arises due to its short life expectancy and the use of low-quality materials, which encourages consumers to replace rather than repair it. According to the E-Stewardship Model published by the Department of Climate Change, Energy, the Environment and Water, a lifespan extension of 10% reduces GHG emissions by 6% in Australia [11]. Hence, promoting repair and reuse is critical in lowering small-WEEE generation rates in Australia.

### 5.4. Recycling and Recovery

Even though repair options are available for Australians, after WEEE collection, recycling is the most preferred option, leading to the recovery of valuable materials. There are two stages of WEEE recycling in the Australian context. First-stage recyclers are engaged in the initial processing, which includes manual dismantling, shredding, magnetic separation, and other pretreatment processes, leading to a lower recycling efficiency, as shown in Figure 4 [104]. Since 90% of the costs are allocated to labour and due to insufficient volumes [110], it is economically viable to outsource both the first stage and downstream recycling overseas. Lack of domestic application for recovered metals and scale for establishing entire recovery operations in the country are two other factors that hinder the recycling value chain [99]. Hence, given proper legislation's nonexistence [111], Australia tends to export WEEE.

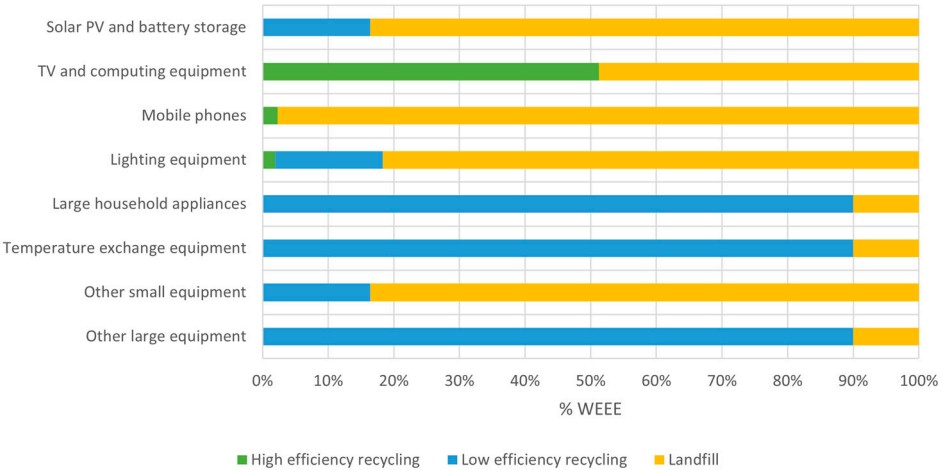

**Figure 4.** Fate of WEEE in Australia [11].

After the first stage of recycling, the material is forwarded to downstream recyclers, mostly carried out overseas [96]. In Australia, scrap computers are the only EEE identified that can be recycled domestically without subsidy. Downstream recycling is preferable due to reduced commodity prices and low international shipping rates [108]. Exports of WEEE fractions after the first stage of recycling reject the potential benefits of downstream industry, which can be gained by recovering precious metals [104]. In terms of waste categories, small WEEE has the highest material content (Figure 5), indicating the need for local material recovery to capture the economic benefits.

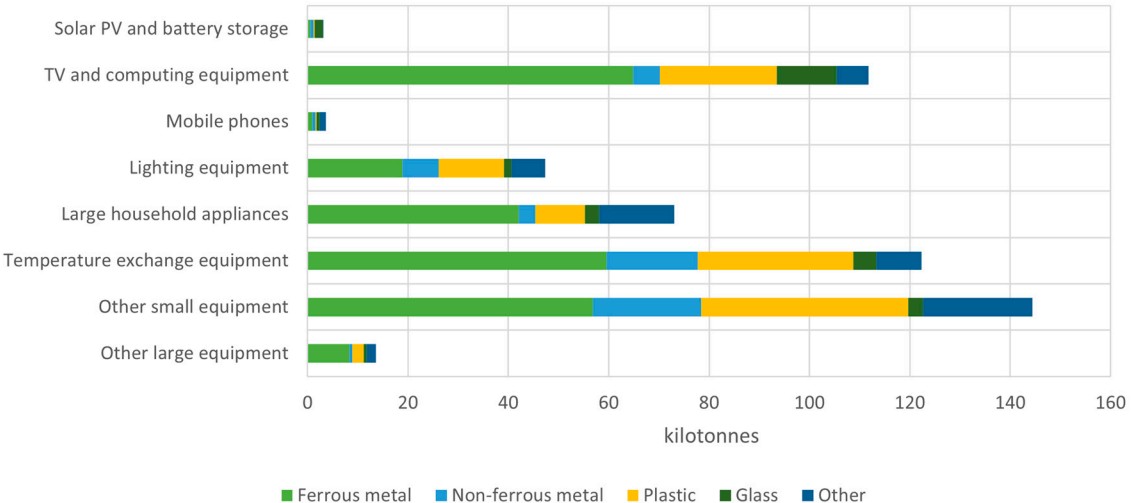

**Figure 5.** Material breakdown at end of life (2019) [11].

Given this context, Australia has two options. The first is to invest in monitoring to minimise improper handling and maximise the safety of the associated labour force. Independent auditors can be recruited by the related parties to ensure this. The second option is gradually investing in downstream recycling to ban WEEE exports [100]. This will benefit the Australian economy by creating more job opportunities while extracting precious metals, leading to high carbon recovery [11]. Establishing state-based central hubs to reduce logistic costs [112] and using modern technology to shift away from labour-intensive recycling are other recommendations observed in the literature to improve WEEE recycling in Australia [94]. Accurate modelling of recycling targets is critical in this case [100], as initial recycling targets for NTCRS resulted in job losses, and due to this, they were amended. The liable parties do not pay off the PROs if they over-collect, leading to the removal of collection points from respective locations. Due to this, certain councils have had to direct the WEEE towards the landfill in the past [7].

However, most small EEE belongs to something other than a specific recycling scheme. In addition, a study conducted for Western Australia indicated that a minor % of small WEEE, 3%, is being sent to recyclers [113]. Even though small WEEE has a larger volume, similar material consumption, and equal potential for hazardous substances, they have a meagre recycling rate in Australia [13] with only 5% of the recoverable value. However, due to the high ferrous metal content, the recycling of small WEEE is mainly driven by the base metals [104].

### 5.5. Landfilling

It is assumed that most small WEEE ends up in landfill [48]. The federal government's ambition to treat and recover every device with a plug or battery in the 21st century indicates that sooner or later a WEEE landfill ban is due [13]. The Victorian government has already followed this effective 2019, and funds were allocated to upgrade WEEE collection and storage facilities across the state [114], which will be followed soon by the State Government of Western Australia (WA) [115]. Avoiding small WEEE from landfill is critical

as hazardous substances like PVC and toxic metals like arsenic, cadmium, lead, and mercury are high in small WEEE, as shown in Figure 6. If the current landfills stop accepting WEEE and recycling mechanisms are not in order, stockpiling WEEE in a dedicated conditioned landfill could also be a viable option to prevent dangerous illegal processing [10].

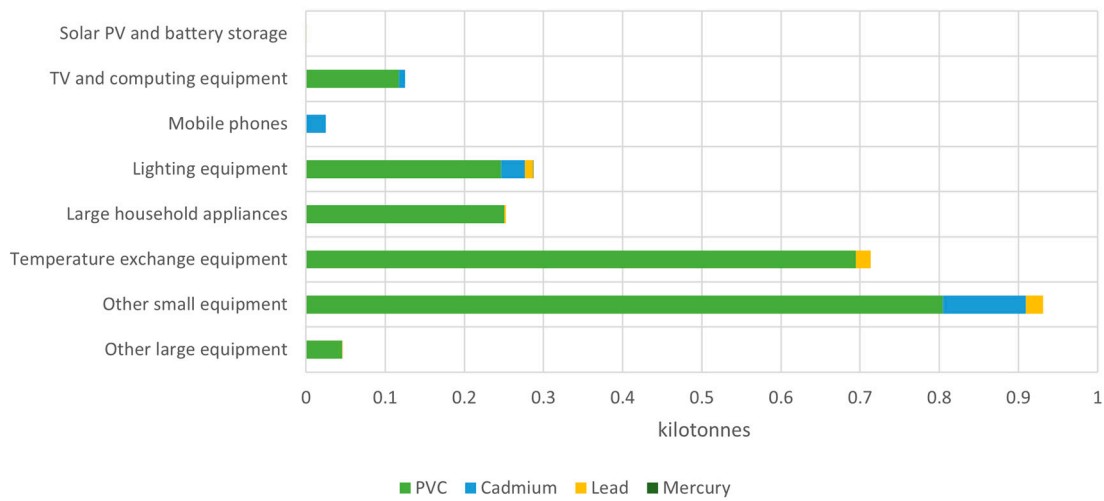

**Figure 6.** Toxic components at end of life (2019) [11].

## 6. Progress of Small WEEE Management in Australia

The Minister of Environment's priority list mentions WEEE, highlighting the need to reduce the amount of WEEE ending up in landfill. Improvement of product designs for circularity has been identified as a course of action to rectify this issue. In parallel with the landfill bans in some states, the government's export ban on mixed plastics will force waste management professionals to implement more stringent procedures for managing WEEE. Considering these facts, recently, the Department of Agriculture, Water and the Environment published a discussion paper titled "Stewardship for Consumer and other Electrical and Electronic Products" to identify the barriers, challenges, and opportunities to develop product stewardship arrangements for EEE that are not covered by NTCRS [13]. The authors analysed this paper and responses received from various stakeholders, which are available publicly. Information identified related to small WEEE is summarised in Table 9.

**Table 9.** Summary of observations related to small WEEE in the discussion paper.

| Options Considered | Observations |
| --- | --- |
| Expansion of NTCRS to include small EEE | Promoting brand-specific collection schemes, divisions of responsibilities, community recycling, drop-off centres, and doorstep recycling is required to maintain efficiency and increase recovery rates. |
| Introducing a container deposit scheme (CDS) | Due to the bulky nature and the dimension variation, pre-processing will be expensive due to the requirement of advanced equipment. If established, resource pooling mechanisms and shared responsibilities are vital to reduce management costs. |
| Providing households with an easily identifiable bag to place small WEEE | It may work with more infrastructure and active participating programmes. Still, it could also contaminate the recycling stream for material recovery facilities (MRFs) while increasing the health and safety risks for both trucks and workers. This will encourage more disposal, reducing the opportunity to reuse the items. |

Following the discussion paper, the Department of Climate Change, Energy, the Environment and Water released a draft regulation for small electric products and solar photovoltaic system waste [103]. The proposed scheme was designed to maximise the

direct collection of small WEEE via free disposal services to households and small-business owners, as most will soon face the effects of a WEEE landfill ban. The costs of the schemes would be passed to the consumers via a small fee, which slightly higher prices will indicate.

The scheme is expected to cover the EEE which is found in households and small businesses weighing up to 20 kg. This includes mobile phones and all other products currently covered by NTCRS. In addition, equipment with embedded batteries is also included in the scheme. However, EoL medical devices, which are considered a biohazard; smoke alarms containing radioactive material; and emerging streams like electronic smoking devices are excluded from the scheme.

The proposed scheme (Figure 7) comprises a single scheme administrator responsible for the scheme outcomes and multiple network operators managing small-WEEE collection, transportation, and recycling services in a specific area. The manufacturers or importers are liable parties who would fund the scheme by paying the administrator a fee related to their quantity of imports.

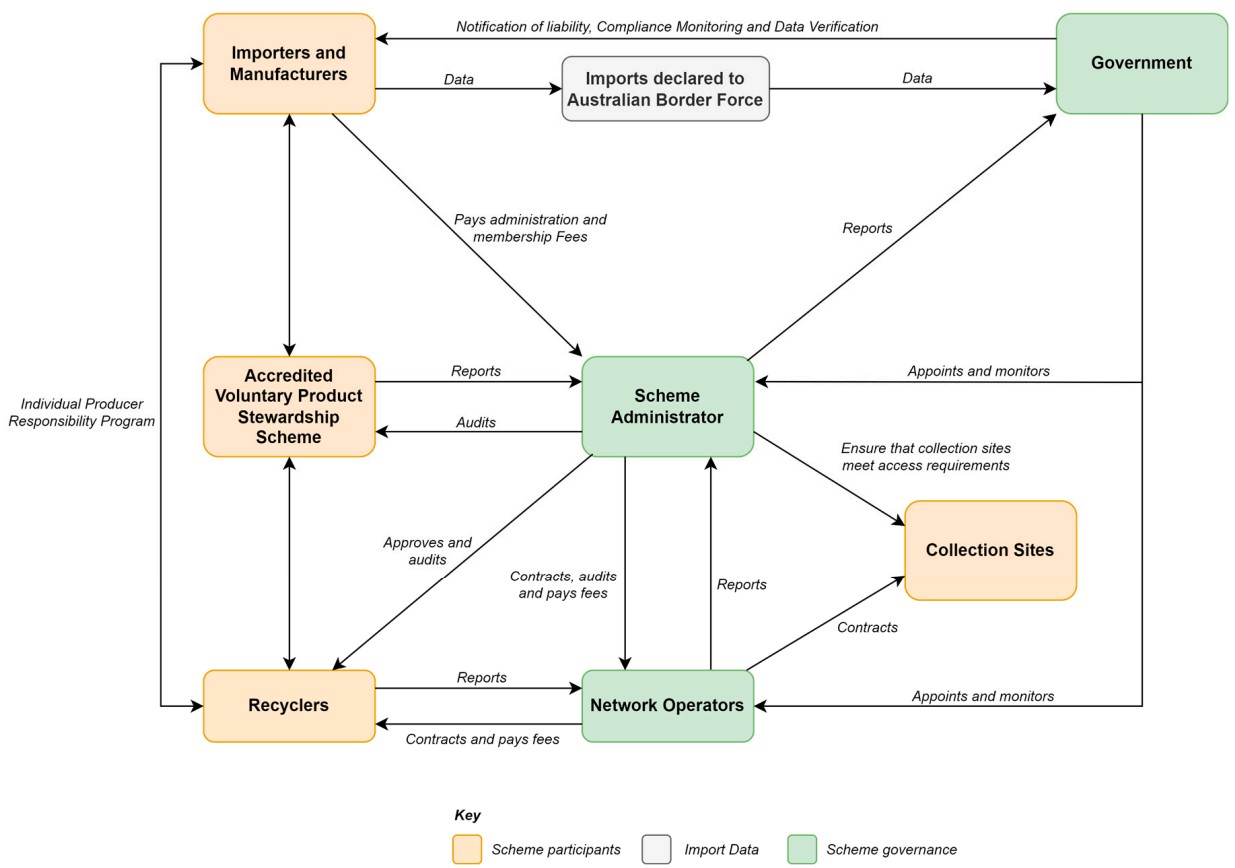

**Figure 7.** Proposed scheme administrative structure [103].

There are five targets and obligations which are proposed in the scheme:

1.  Calculation of WEEE recycling rates;
2.  Information on the ratio of recovered materials for remanufacturing purposes;
3.  Implementation of accessible drop-off centres;
4.  Encouraging the maximisation of reuse of EEE;
5.  Carry out awareness campaigns encouraging CE practices like repair and reuse.

The scheme target is the minimum amount of small WEEE that should enter the recycling system each year. The percentage target is expected to be increased annually. A single target for the entire scheme is suggested, as separate targets would add additional administrative burdens. A recovery target is also proposed to trace the recovered materials'

quantity, type, origin, and destination. Even though valuable materials are expected to be kept in circulation, plastics with POPs in small WEEE can be incinerated for energy recovery.

Reasonable access to the collection services is expected to be provided in all areas, including retail shops, standalone sites, council-hosted sites, popup events and sometimes postback mechanisms for relevant products. These collection sites should follow appropriate standards while promoting public awareness. Self-nominated collection sites can also be recognised to enhance the collection mechanisms, provided that the operation of such sites is not funded via the scheme.

This scheme promotes repair and reuse via educational and awareness campaigns rather than imposing a target. It is expected that separate collection mechanisms will be made available at collection sites as relevant organisations request them. However, measures will be taken to avoid cherry-picking high-value WEEE for recycling. The scheme administrator should keep track of these diversion rates. Recognition for other recycling method arrangements like direct recycling by the importer itself and voluntary schemes should be given by absolving their liabilities to the system. A reasonable base fee is expected to be charged by the liable parties if they reduce the sensitive weight to zero.

## 7. Future Directions for Australia

The sections above derived significant insights to manage small WEEE in Australia effectively. Numerous barriers and challenges became evident among both consumers and stakeholders. Consumers faced issues such as hoarding due to negligence and emotional attachments, lack of knowledge about proper disposal and repair, hygiene concerns when reusing items, and high costs of repairs. On the other hand, WEEE management professionals encountered challenges like insufficient infrastructure to handle the diversity of small WEEE, unappealing weight-based collection targets, and an overemphasis on recycling rather than other aspects. Achieving a synergy between consumer expectations and regulatory requirements is essential when managing a delicate waste stream like small EEE. With this in mind, future recommendations were provided, centring around a dedicated stewardship scheme, EPR, improved infrastructure, and changes in consumer behaviour.

### 7.1. Dedicated Stewardship Scheme

Managing small WEEE is always challenging due to the wide variety of products. Japanese SHARL covering 28 different products indicates that a dedicated stewardship scheme is of utmost requirement. In the case of Australia, it can only to cover the products in Table 1, or existing systems can be combined to cover a broader range of products, as mentioned above. Since NTCRS is familiar to customers and the associated PROs have a higher success rate, the latter is preferred as it would only require upgrading infrastructure, technologies, and administrative capacities. Given the higher number of retailers for small EEE in Australia, this would be an ideal approach that suitable takeback and postback schemes, doorstep approaches, and brand-specific collection schemes should strongly support. However, unlike the current scenario, local authorities should be considered significant stakeholders as they should be provided additional funds to maintain the operations swiftly. The scheme should carry a digital database to ensure all retailers, online sellers, and drop shippers are involved in small-WEEE management.

Performance of the scheme can be based on the recycling and recovery targets, which should consider both the quality and quantity of recycled materials. Even though the repair and reuse culture for small WEEE is insignificant, it should be noticed, and related targets can be incorporated along the way after obtaining reliable data on the product lifespans.

A simple yet realistic fee structure should enable the smooth functioning of the scheme. Apart from imposing a fee based on EEE put on the market (PoM) weights, modulation is also necessary to encourage more product circularity. As carried out in France and Italy, advanced modulation can be established, which includes bonuses and maluses [116]. This should be performed considering social and economic aspects, as it should not incur extra administrative burdens or complexity to the scheme. Visibility of fees can be direct, which

may account for at least 10% of the product cost, or indirect via a product star rating to ensure consumer transparency [117]. However, in-depth research should be conducted for Australia before applying these fees to small WEEE, considering the complexity of the products and the market.

### 7.2. EPR to Increase Product Circularity

Currently, in Australia, EPR is focused mainly on recycling and recovery, giving less attention to repair and reuse, which are at upper levels in the waste hierarchy. Hence, to ensure higher product circularity, regulations should focus on promoting lifespan extensions by incorporating modular design concepts. This is also crucial in the recycling stage to ensure the ease of disassembly, reducing recycling costs. Eco-designs are also encouraged to use less hazardous materials. Since most of the small EEE PoM in Australia consists of imports, indicators like the repairability index and reusability potential can be used to encourage importing repairable and reusable small EEE. Copyright laws should also be amended to share the repair information with the repair shops and customers via labelling schemes [6] or embedded repair codes. Producers should also allow repair shops to use interoperable spare parts in their small EEE, reducing the cost of spare parts. Local repair shops should be given the fullest support by providing adequate resources. Since around 40% of the repair cost is associated with labour and other expenses [118], consumers can be empowered with self-repair manuals to encourage Do It Yourself (DIY) repairs, reducing repair costs. In addition, product warranties given by retailers should cover repairing from different agents rather than sticking to the manufacturer's authorised agents, which usually take a long time to process. Continuous firmware updates are also necessary rather than halting them to force customers to buy new alternatives. These suggestions are also confirmed by the recommendations of the recent Right to Repair report published by the Australian Productivity Commission [6]. Among all these, the release of products to the market should be carried out scientifically, avoiding early obsolescence of functional small EEE.

### 7.3. Customers' Responsible Consumption

While taking all the measures to reduce waste generation and increase treatment from the producer's end, customers also have a crucial role. At the point of purchase, they should always check on the quality, durability, upgradability, and repair options targeting a more prolonged product use. In addition, they should be familiar with the user manuals, software updates, and safe handling to ensure a longer product lifetime. Including eco-labels and a visible fee for waste management on the product will make customers aware of how important it is to consume responsibly [119]. While informing customers during the point of sale, dedicated awareness programmes are also needed to encourage a repair and reuse culture and to avoid stockpiling and disposing of the small WEEE in the wrong stream. Consumers should also be encouraged to provide frequent feedback on the forward and reverse supply chains to improve the system.

### 7.4. Expansion of Collection, Treatment, and Recovery Infrastructure

To expand the collection infrastructure, retailer shops and supermarkets should be allocated a separate collection space with storage facilities for small WEEE. Mobile collection units and postback schemes can be introduced to increase regional access. Looking at recycling, the current first-stage recycling infrastructure should be enhanced to cater for small WEEE, which has a broader range of products. Then, under a proper regulatory framework, downstream recycling should be well established. PROs should follow strict rules to monitor the recovery rates. While sorting and dismantling should be performed in decentralised locations, central recycling plants should be used for further treatment and material recovery.

In conclusion, due to the complexity of small EEE, industry stakeholders, regulatory bodies, and consumers all play an essential task in managing small WEEE. While the authorities take necessary actions to increase EPR and ensure convenient collection and

treatment of small WEEE, consumers should use small EEE responsibly, opting for repair and reuse over immediate disposal after functional use. Hence a holistic approach is required to manage small WEEE sustainably and efficiently in Australia.

**Author Contributions:** Conceptualisation, G.J.; methodology, G.J.; validation, S.H. and P.K.; formal analysis, G.J.; investigation, G.J.; resources, G.J., S.H. and P.K.; data curation, G.J.; writing—original draft preparation, G.J.; writing—review and editing, S.H. and P.K.; visualisation, G.J.; supervision, S.H. and P.K. All authors have read and agreed to the published version of the manuscript.

**Funding:** This research received no external funding.

**Institutional Review Board Statement:** Not applicable.

**Informed Consent Statement:** Not applicable.

**Data Availability Statement:** Data are contained within the article.

**Acknowledgments:** This research was conducted with funding received by G.J. through the Griffith University Postgraduate Research Scholarship (GUPRS) and Griffith University International Postgraduate Research Scholarship (GUIPRS), both offered by Griffith University, Australia.

**Conflicts of Interest:** The authors declare no conflict of interest.

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
