# Peer review of "Management of Small WEEE: Future Directions for Australia"

_sustainability, doi:10.3390/su151813543_

Round 1

Reviewer 1 Report

This paper describes future directions for the generation of small Waste Electrical and Electronic Equipment (WEEE) which is rapidly increasing due to the diverse eletronic products. Australia being one of the largest WEEE generator must incorporate clear policies for recycling electronic wastes before their harmful contents proliferate in the land.

The climate impact of WEEE must be elaborated in the paper. Because the production and disposal of electronic devices contribute to carbon emissions and climate change. Responsible recycling and proper disposal can help reduce the carbon footprint associated with electronic waste. 

One figure should demonstrate the composition of WEEE in Australia to depict the percentage of harmful contents inside them, e.g. lead, mercury, cadmium, brominated flame retardants, and PVC etc. It is important to note that the piles of electronic waste can also disrupt local ecosystems and wildlife habitats.

Line 122: Is the sentence complete? If yes, 'small' should be written as 'Small'.

A part from methods mentioned in the paper, following recommendations are noteworthy:

1) In some countries, the manufacturer take-back programs are also effective which encourage electronics manufacturers to establish take-back programs where they accept their products for recycling at the end of their life cycle.

2) Installing E-Waste bins in the country which are dedicated e-waste collection bins in convenient locations such as shopping centers, community centers, and office buildings is also effective.

3) Encourage repair and refurbishment of small electronic devices to extend their lifespan. Support local repair businesses or provide resources for DIY repairs.

4) Deploy mobile units equipped to collect and transport small WEEE from different locations, particularly in areas where access to recycling facilities is limited.

5) Encouraging electronics manufacturers to design products with recyclability in mind i.e. 'Green Design', using eco-friendly materials and minimizing the use of hazardous substances may have higher impact.

It's important for Australia to adopt a holistic approach to electronic waste management, involving government agencies, industry stakeholders, and the public, to create a sustainable and effective system for handling small WEEE in the future.

Overall, the paper's quality is good. References are recent and the research is well designed to promote policy steps in the management of WEEE.

Quality of English is good except few typos.

Reviewer 2 Report

This manuscript presents an analysis of the management and prospective path of small WEEE (Waste Electrical and Electronic Equipment) in Australia. Nevertheless, the document's general arrangement seems to lack efficient organization, and the absence of results is striking. To enhance the document's quality, it is suggested that the authors consider restructuring the content to attain better coherence.

Global WEEE Management

The management of this section fell short, as it exhibited storytelling elements that were not suitable. Although the authors introduced a range of European countries at the outset of subsection 4.1, the subsequent sections (4.2 and 4.3) exclusively centered around Japan and Canada. To foster a more comprehensive perspective, I recommend that the authors maintain balance by delving into discussions about regions like Europe, Asia, and America.

Results:

The absence of results and discussion is notable. It is recommended that the authors include an additional paragraph to present the significant insights derived from this study. Furthermore, there are suggestions to enhance the strategy and raise social awareness regarding the utilization of small WEEE in Australia that need to be established. The authors should also provide a comparison between the current regulations and the recommendations proposed in this study, aiming to enhance social awareness and formulate appropriate policies for small WEEE in the country.

Others mistake:       

- Several typos are present within this manuscript. To illustrate, the word "keyes" in line 101 and "Thes" in line 688 require correction. Kindly proceed with the essential revisions.

- Numerous abbreviations are used without their corresponding full words being initially stated.

There are many typoes.

Please carefully recheck and revise.

Reviewer 3 Report

The research topic is meaningful and interesting, but the authors should clear indicate the main contributions of this study.

1. Abstract. What do white and grey literature mean? What are online databases? A clear definition is recommended.

2. Background. The research topic of this manuscript is small WEEE, but the background content is lengthy and the focus is not prominent. It is suggested to discuss the topic with a focus on small WEEE.

3. Small Waste Electrical and Electronic Equipment (small WEEE). What are the criteria for classifying electronic products in Table 1? How can we determine whether a certain electronic product belongs to small-scale electronic products rather than large-scale electronic products? Is it based on weight or size?

4. Materials and Methods. Literature searches were conducted in multiple databases. How many articles were retrieved in total? How many of them are directly relevant? Which journals are these articles primarily concentrated in? It is recommended to provide this information.

5. Global WEEE Management. Table 3 presents the barriers and challenges of electronic waste recycling, not specifically related to small WEEE. If the barriers to recycling small WEEE are the same as those for large WEEE, then the significance of studying small WEEE recycling issues may seem limited. Can the information in Table 3 be updated to focus on small WEEE?

6. WEEE Management in Australia. This section is a summary of existing literature, and it is lengthy. It is recommended to streamline it appropriately.

Round 2

Reviewer 2 Report

All recommendations provided by the reviewers have been implemented, resulting in an enhanced version of the manuscript. The authors have diligently revised the manuscript, addressing both the reviewer's comments and rectifying any minor errors. Nevertheless, it is advisable to address a small issue before proceeding with the publication process.

 - "Figure 2: Within Figure 2, two bar charts are presented without accompanying captions for identification. To enhance the clarity of this figure, the authors are advised to provide subfigure captions, such as Figure 2a and Figure 2b."

Author Response

Dear Reviewer,

Authors accept the comment, and corrections are made